# Visual Performance After Bilateral Implantation of a New Enhanced Monofocal Hydrophobic Acrylic Intraocular Lens Targeted for Mini-Monovision

**DOI:** 10.3390/life15010064

**Published:** 2025-01-07

**Authors:** Hugo A. Scarfone, Emilia C. Rodríguez, Jerónimo Riera, Maira Rufiner, Martín Charles

**Affiliations:** 1Clínica de Ojos de Tandil, Av. Santamarina 760, Tandil B7000, Argentina; rodriguezemiliacarolina@gmail.com (E.C.R.); riera.jeronimo1986@gmail.com (J.R.); mairaguadaluperufiner@gmail.com (M.R.); 2Centro Oftalmológico Charles, Buenos Aiers C1116, Argentina; doccharles@gmail.com

**Keywords:** intraocular lens, IOLs, monofocal IOLs, cataract surgery, Clareon, intermediate vision

## Abstract

Background: The aim of this study was to evaluate visual outcomes and patient satisfaction after bilateral implantation of a new hydrophobic acrylic intraocular lens called Clareon (Alcon) using the mini-monovision technique. Methods: A single-center, prospective, nonrandomized study was conducted in Tandil (Buenos Aires, Argentina), including patients scheduled for cataract surgery. To achieve mini-monovision, the spherical equivalent was calculated between −0.25 and +0.25 D for the dominant eye, and between −0.75 and −1.00 D for the non-dominant eye. The main outcomes were uncorrected distance visual acuity (UDVA) and uncorrected intermediate visual acuity (UIVA) evaluated at 66 cm. A secondary outcome, patient satisfaction, was assessed using the CatQuest-9SF questionnaire. Results: The mean binocular UDVA was 0.01 ± 0.05 logMAR three months after surgery, while the mean binocular UIVA was 0.20 ± 0.06 logMAR. The postoperative mean spherical equivalent in the dominant eye was −0.27 ± 0.12, and in the non-dominant eye was −0.87 ± 0.25. Before surgery, the CatQuest-9SF questionnaire revealed that 83.33% of patients were dissatisfied with their vision during daily activities. Over 50% reported significant difficulties with reading newspapers, sewing, and reading TV subtitles. Additionally, 66.6% struggled with recognizing faces, 50% with seeing product prices, and 50% with walking on uneven ground. Post-surgery, most patients experienced improved vision for daily tasks, with no reports of high dissatisfaction or significant difficulties. Patients were quite satisfied with their vision for hobbies and TV subtitles, and very satisfied (90%) with seeing supermarket prices. Conclusions: patients implanted with a new enhanced monofocal IOL using the mini-monovision technique showed improved distance and intermediate visual acuity, reduced need for glasses, and expressed a high degree of satisfaction.

## 1. Introduction

The primary goal of cataract surgery has shifted from merely restoring blurred vision at far distances to improving vision at all functional distances with total spectacle independence. Achieving pseudo-accommodative multifocal vision usually sacrifices sharpness for depth of field [1]. Despite advancements in intraocular lens (IOL) designs to meet higher patient expectations, monofocal IOLs remain the most commonly implanted due to their lower cost, excellent single-focus vision outcomes, and low incidence of photic phenomena such as halos and glare [2]. Monofocal IOLs are also suitable for patients with corneal or macular diseases where multifocal IOLs are not recommended. However, monofocal IOLs are limited to restoring either distance or near vision. As intermediate visual acuity becomes more important in daily life, new optical designs in monofocal IOLs aim to provide the same distance visual acuity as standard monofocal IOLs while improving intermediate vision without causing dysphotopsias [3].

Monofocal IOLs can provide a slightly extended depth of focus (DoF) depending on their optical design, although this has not been widely discussed historically [4]. New bilateral implantation techniques, such as pseudophakic mini-monovision, induce controlled anisometropia to create pseudo-accommodation in presbyopic patients, offering a wider range of vision [4,5]. In monovision, the dominant eye is corrected for distance vision and the non-dominant eye for near to intermediate vision, with intended residual myopia ranging from −0.75 to −1.75 diopters. Studies have shown satisfactory spectacle independence following mini-monovision with monofocal IOLs [3].

The Clareon IOL (SY60WF) by Alcon Laboratories (Forth Worth, TX, USA) is a new intraocular lens made from AcrySof hydrophobic polymer, with a slightly higher water content of 1.5%. It features an improved milled edge profile and quality, and a manufacturing process that minimizes surface roughness and material inconsistencies. The Clareon IOL shares many mechanical properties with the AcrySof IOL, including a 6.0 mm diameter asymmetric biconvex posterior optic, an anterior aspheric surface (−0.2 μm), an overall length of 13.0 mm, and a similar haptic design, allowing it to maintain its position within the capsular bag across various sizes [6].

The Clareon IOL (SY60WF) is preferred for excellent quality of vision and distance visual acuity, with reports indicating some improvement in intermediate visual acuity [3,7,8]. This study aimed to assess visual outcomes for far and intermediate distances, refractive results, defocus curves, and patient satisfaction in individuals who underwent bilateral implantation of the Clareon IOL using the mini-monovision technique to enhance intermediate vision.

## 2. Materials and Methods

### 2.1. Study Design and Bioehtics

A single-center, prospective, nonrandomized study was conducted at the Tandil Eye Clinic, Tandil, Buenos Aires, Argentina, following the principles of the Declaration of Helsinki. Patients provided informed consent after receiving an explanation of the research and its intent. This study was approved by the Research Ethics Committee of the Argentine Society of Ophthalmology (CEISAO; registration number 10450).

### 2.2. Population, Parameters, and Follow-Up

The study included patients of both genders, aged 50 years or older, who had signed an informed consent. Patients with healthy eyes and cataracts were included. Patients who had undergone previous eye surgeries, such as refractive surgeries, glaucoma surgeries, and vitreoretinal surgery, were excluded from the study. Additionally, patients with a history of ocular pathologies, such as glaucoma, uveitis, retinopathy, macular lesions, pupillary anomalies, and corneal dystrophies, as well as severe dry eye, were excluded. Patients who failed to achieve the target spherical equivalent for the dominant eye within a range of −0.25 D to +0.25 D, and for the non-dominant eye within a range of −0.75 D to −1.00 D, were excluded from the study. Additionally, patients needing a cylindrical correction exceeding 0.50 D due to corneal astigmatism were also excluded.

Patients who agreed to participate were monitored preoperatively with visual acuity with and without correction using a Tomey TMS-4 topographer (Tomey; Phoenix, AZ, USA), ARGOS optical biometer (Alcon; Forth Worth, TX, USA), and TOPCON KR-800 automatic refractometry keratometer (Topcon corporation; Itabashi, Tokyo, Japan). Pupil diameter and spherical aberration of the cornea (Q factor) were evaluated using an Eyestar 900 Haag Streit optical biometer (Haag Streit Groups; Schweiz, Germany). The retina was evaluated by ultra-high-speed full-range OCT (Solix-Optovue; Fremont, CA, USA) and an Optos California (Opctos Inc., Marlborough, MA, USA). The endothelial cell count was determined by specular microscopy (TOMEY EM 4000; Tomry; Phoenix, AZ, USA). Slit lamp biomicroscopy, Goldmann applanation tonometry, and dilated fundoscopy were performed. Study of the ocular surface was performed under slit lamp biomicroscopy and an SBM Sistemi model IDRA (Orbassano, TO, Italy).

An ocular dominance test was performed using a Porta Test in the first consultation. To do this, the patient must extend their arms and then, with both eyes open, align a finger or a pencil with a distant object. They must fix their eyes on it and alternately close one or the other eye. The dominant eye will be the one in which the pencil is more aligned with the object. To obtain mini-monovision, an Argos was used for preoperative measurements and calculations (Barret Universal II Formula was optimized for axial length measurement by sum of segments). The dominant eye was corrected for distance vision, while the non-dominant eye was corrected for intermediate vision. The spherical equivalent was calculated between −0.25 and +0.25 D for the dominant eye, and between −0.75 and −1.00 D for the non-dominant eye (the non-dominant eye was operated on first, and then the dominant eye one week later).

Uncorrected binocular distance visual acuity (UDVA) and corrected binocular distance visual acuity (CDVA) were assessed at 4 m. Uncorrected binocular intermediate visual acuity (UIVA) was assessed at 66 cm. The degree of patient satisfaction with distance and intermediate vision was evaluated using a questionnaire (Catquest-9SF) [9] that was completed preoperatively and after bilateral Clareon implantation with the mini-monovision technique at 3 months. The Catquest-9SF test was performed without glasses. Main outcomes were evaluated at 90 days, including visual performance (visual acuity, spherical equivalent, and defocus curve) and patient satisfaction (Catquest-9SF).

### 2.3. Surgical Technique

Regarding the surgical technique, the patient was dilated with three drops of tropicamide-phenylephrine every 5 min for half an hour before surgery. Topical anesthesia was performed with eye drops (Proparacaine). Phacoemulsification was performed with a Centurion Vision System (Alcon; Forth Worth, TX, USA), with Active Fluidics and an Active Sentry handpiece using a low-pressure setting (IOP 30 mmHg, flow rate 35 cc/min, and vacuum 350 mmHg). Surgeries were performed on both eyes of each patient with a difference of 7 days between surgeries. All surgeries were performed by the same surgeon, Hugo Scarfone (HS).

### 2.4. Statistics

Data were analyzed using Statsdirect statistical software, Version 2.7, and a Mann–Whitney test was used to compare numerical variables. The level of significance was considered at *p* < 0.05. The minimum sample size consisted of 30 patients (60 eyes) undergoing cataract phacoemulsification surgery with intraocular lens implantation (Clareon^®^ monofocal non-toric intraocular lens). Given that visual performance was the primary parameter to be evaluated, a statistical power of 80% and a 95% confidence interval were used to calculate the sample size, assuming a standard deviation of 0.5 and a margin of error of 0.1.

The dataset of the study is available in this public repository: https://zenodo.org/records/13716481.

## 3. Results

### 3.1. Demographics

Of the 36 patients with bilateral Clareon implants, only 30 were included (21 women and 9 men) because the refractive target was not reached in 6 patients. Complete preoperative demographic characteristics are summarized in Table 1, including axial length, pupil diameter, spherical aberration of the cornea (Q factor), and IOL power.

### 3.2. Visual Performance

The mean binocular UDVA (4 m) was 0.01 ± 0.05 logMAR three months after surgery, while the mean binocular UIVA (66 cm) was 0.20 ± 0.06 logMAR.

The postoperative mean spherical equivalent in the dominant eye was −0.27 ± 0.12 and in the non-dominant eye was −0.87 ± 0.25. The visual acuity results are summarized in Table 2, and the distribution of percentages of eyes within a given visual acuity is summarized in Table 3.

The defocus curve (Figure 1) showed a peak that corresponds to the best visual acuity of 0.00 D (4 m); subsequently, a reduction in visual acuity is observed with a gradual negative defocus. A soft and broad profile is observed along the entire curve toward the myopic range, especially within the defocus range that corresponds to intermediate vision (approximately—1.50 D of defocus, corresponding to 66 cm).

### 3.3. Patient Satisfaction

The results of the CatQuest-9SF questionnaire obtained in the third month after surgery are shown in Table 4. Before surgery, 83.33% of the population reported being dissatisfied with their vision when performing daily activities. The greatest difficulties were observed when reading the newspaper, performing sewing tasks, or reading subtitles on a television, with more than 50% of the patients being very dissatisfied. For other activities such as recognizing people’s faces (66.6%), seeing product prices (50%), or walking on uneven ground (50%), patients were quite dissatisfied. Most patients reported that their vision improved after cataract surgery in various activities of daily living. No patient reported being very dissatisfied or having great difficulties after surgery. Patients reported being quite satisfied when carrying out their favorite hobby or when reading subtitles on a television, and very satisfied in 90% of cases when seeing supermarket prices.

## 4. Discussion

The present investigation evaluated the results of cataract surgery with implantation of a Clareon monofocal aspheric IOL calculated for mini-monovision in 30 patients. In our study, the Clareon monofocal IOL with the mini-monovision technique significantly improved patients’ uncorrected distance and intermediate visual acuity and reduced their dependence on spectacles. Most patients reported little to no use of spectacles for daily activities like computer work, distance viewing, and general tasks after the procedure. While a few patients still required spectacles for reading and night driving, this is a common issue seen with monofocal and sometimes with EDOF or multifocal intraocular lenses [10]. In this sense, we consider the better potential for excellent best-corrected visual acuity as an advantage of the monofocal lens [1]. The UDVA results of our study were superior to those reported by other investigations evaluating “plus” monofocal lenses [3] and EDOF IOLs [11] used with the mini-monovision technique, while the intermediate visual acuity outcomes were comparable to those studies.

It is worth noting that patients increasingly seek good intermediate vision due to the widespread use of tablets and cell phones in both work and recreational settings [12]. Our research also analyzed the binocular defocus curve, revealing a gentle slope and a gradual decrease in visual acuity as defocus increases. It is important to note that in the binocular defocus curves from our study, we found a greater decrease in UDVA when the defocus was positive. The average UDVA for a defocus of +1 D was 0.39 logMAR (±0.08), whereas for a defocus of −1 D, the average UDVA was 0.25 logMAR (±0.06). This difference was statistically significant. Based on this finding, we recommend attempting IOL power calculations for the monofocal Clareon to achieve a target of −0.25 D.

Visual acuity of 0.37 logMAR (±0.04) with an equivalent of −1.5 D was achieved at 66 cm distance, similar to a curve found by Nikola Tomagova et al., with an extended range lens calculated for mini-monovision [11]. Stodulka et al. [13] assessed the visual performance of an EDOF IOL and discovered that a visual acuity of 0.4 logMAR was attained with a defocus of −1.50 D, which is equivalent to a distance of 66 cm. In our own defocus curve, we achieved a visual acuity of 0.37 logMAR for the same degree of defocus.

When implanting EDOF IOLs, some unwanted visual phenomena have been described in many publications, including reduced vision quality and halos. According to a systematic review by Yining Guo and Yinhao Wang, halos were reported as the most frequent, severe, and bothersome visual symptom [14]. Weber et al. described glare as a disabling symptom in some patients with EDOF IOL implantation [15]. Goldberg et al. have demonstrated that implanting monofocal IOLs (AcrySof^®^ IQ IOL or AcrySof^®^ Toric/IQ IOL) using the mini-monovision technique significantly enhanced patients’ uncorrected binocular visual acuity (0.09 ± 0.09 logMAR) and reduced their reliance on glasses in over 90% of cases [16].

Although Clareon is a lens that has many similarities to the AcrySof IQ platform, it differs mainly because it is designed with hydrophobic acrylic that incorporates hydroxyethyl methacrylate (HEMA). Because of these characteristics, it has a slightly higher water content than AcrySof (1.5% compared to 0.4%, respectively), which gives better physical-optical properties, resulting in greater clarity, according to a study conducted by Werner et al. [17]. Another difference is the design of the edges. This is another factor that would give better optical performance to Clareon, in addition to decreasing the development of posterior capsular opacification as reported by Nuijts et al. in a three-year multinational study [6].

The induction of spherical aberration (SA) to enhance depth of focus (DoF) should consider the inherent aberration of the eye, which can differ between patients. An IOL inducing a negative SA will produce less DoF in an eye with higher positive corneal SA since the cornea will compensate for the aberration induced by the IOL [18,19]. On the other hand, the required SA will depend on a patient’s pupillary diameter (PD). Younger presbyopic patients have higher PDs, and simulations in this population for 4.5 mm pupils generally agree that an induction of C40 between −0.15 μm and −0.18 μm can induce an extension of the DoF of approximately 0.5 D, with a loss of one line of VA and an associated myopic shift of 0.5 D [19]. Our findings support the idea that there is an increase in depth of focus associated with a negative SA and subsequent improvement in intermediate vision. It is important to note that this performance is affected by the size of the pupil.

Our study has a number of limitations that should be mentioned, one of which is that we have not included contrast sensitivity measurements for different pupil sizes, which would allow us to better understand the performance and function of this lens under different light conditions. Another aspect to be considered is related to the aberrations evaluations, since it would have been interesting to measure the total optical aberrations beyond the corneal aberrations, pre- and postoperatively, in order to know more about the lens behavior and aberrometric variations that occurred. In our protocol, we only considered measuring corneal aberration (Q factor) as preoperative demographic data, but no postoperative measurements were scheduled. But the most relevant limitation is related to the study design, since a control group was not included. The results we have obtained would be stronger if they had been directly compared with patients in whom emmetropia had been preoperatively programmed as a control group. In this way, it would be possible to know the difference between both techniques, using the same lens, in cases operated by the same surgeon. Likewise, the design used for our study was adjusted to a more specific objective, which was to evaluate the visual performance of patients implanted with the Clareon lens and a specific monovision technique. We hope that our work may justify the development of future comparative studies that aim to compare emmetropia and mini-monovision techniques with this lens.

The biggest challenge for monovision techniques is proper patient selection as these techniques are based on neuroadaptation and good blur suppression. The choice of mini-monovision for cataract surgery should consider the patient’s motivations, daily activities, treatment costs, and ability to tolerate potential side effects. The main disadvantage is that some patients may not tolerate mini-monovision and may require near correction for the distance eye for prolonged reading and distance correction for the near eye for certain distance tasks, such as driving in adverse conditions.

In relation to patients’ satisfaction, evaluated by Catquest-9SF, we would like to highlight a relatively controversial aspect. Patients were asked to respond regarding their visual performance without glasses before and three months after surgery. In our series, we found that 66.6% of the patients were very satisfied to perform handicrafts or sewing after surgery. These are visual activities that require great near-vision performance. But the mean binocular near vision of our series for a distance of 40 cm was around 0.7 LogMAR (as seen in the defocus curve), which would not be sufficient to perform this conventional near-visual task. Our interpretation of the results is that the patients may have been able to perform the tasks satisfactorily but by moving away the object to be visualized by up to 60 or 70 cm through stretching their arms. Therefore, we would like to emphasize that although the Catquest is a scientifically validated tool, the information obtained with questionnaires such as the Catquest is still a subjective opinion of the patient, which will also have variables that are difficult to mitigate since he/she may seek to place the object to be visualized in the most comfortable position, and this may not always correlate with the distances that we conventionally consider to be near or intermediate.

Providing patients with comprehensive information about the expected benefits, risks, and costs of each option is crucial to support informed decision-making and patient-centered care. Future research should further assess spectacle dependence during daily activities using emerging technologies [20], as well as compare the long-term efficacy of mini-monovision with monofocal, “plus” monofocal, or EDOF IOLs versus EDOF or multifocal IOLs in randomized controlled trials.

## 5. Conclusions

Our study demonstrates the effectiveness of pseudophakic mini-monovision with a Clareon mono-focal IOL in achieving satisfactory visual outcomes. The results indicate that this approach leads to significant improvements in both distance and intermediate visual acuity. Patients reported high satisfaction levels and a reduced need for spectacles. Specifically, binocular uncorrected distance visual acuity and binocular uncorrected intermediate visual acuity showed substantial enhancements at 3 months. Also, the study highlights the low rate of complications associated with this treatment. This technique may be particularly beneficial for individuals with occupations that involve susceptibility to dysphotopsias or those with retinal disorders as it provides a reliable solution for presbyopia correction using enhanced monofocal IOLs. Additionally, mini-monovision is a lower-cost alternative for patients who cannot afford premium multifocal IOLs but still desire some degree of spectacle independence.

## Figures and Tables

**Figure 1 life-15-00064-f001:**
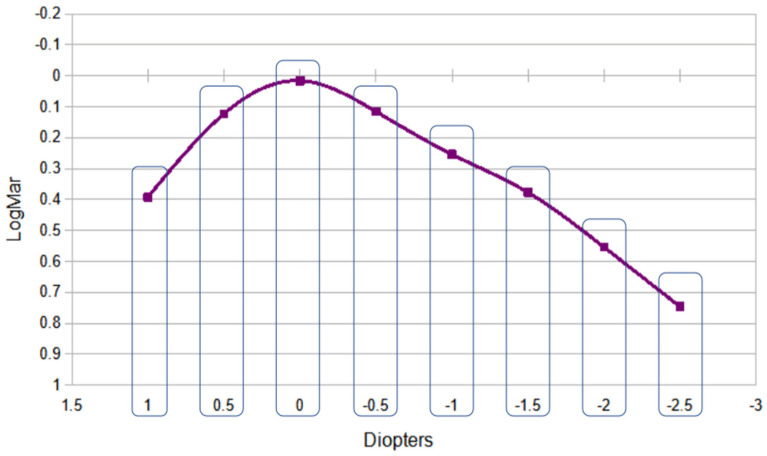
Binocular defocus curve of patients implanted with Clareon intraocular lens and mini-monovision technique.

**Table 1 life-15-00064-t001:** Preoperative demographic characteristics.

Parameters	Values
Age (years)	67.3 ± 7.25 (49–76)
AL (mm)	23.3 ± 0.9 (22.0–24.8)
ACD (mm)	3.25 ± 0.4 (2.6–4.0)
Pupil Diameter (mm)	3.39 ± 0.4 (2.7–3.8)
Q Factor	−0.35 ± 0.1 (−0.5 to −0.15)
IOL Power (D)	22.2 ± 2.2 (18–25)

**Table 2 life-15-00064-t002:** Postoperative visual performance obtained 3 months after surgery. Mean, standard deviation, and range values are presented.

Binocular Visual Acuity	Results (Mean, Standard Deviation, and Range) Presented in LogMAR
UDVA	0.01 ± 0.05 (−0.1 to 0.2)
CDVA	0.007 ± 0.02 (0 to 0.1)
UIVA	0.2 ± 0.06 (0.1 to 0.2)
CDIVA	0.3 ± 0.074 (0 to 0.5)

UDVA: uncorrected distance visual acuity; CDVA: corrected distance visual acuity; UIVA: uncorrected intermediate visual acuity; CDIVA: corrected distance intermediate visual acuity.

**Table 3 life-15-00064-t003:** The distribution of percentages of eyes within a given visual acuity.

Binocular Visual Acuity	Mean ± SD (Min/Max) logMAR	20/15 or Better (−0.1 logMAR)	20/20 or Better (0.0 logMAR)	20/25 or Better (0.1 logMAR)	20/30 or Better (0.2 logMAR)	20/40 or Better (0.3 logMAR)
UDVA (4 m)	0.01 ± 0.05 (0.1 to −0.1)	7.69%	84.62%	100%	100%	100%
UIVA (66 cm)	0.20 ± 0.06 (0.1 to –0.2)	0%	0%	15.38%	76.91%	100%

SD: standard deviation; UDVA: uncorrected distance visual acuity; UIVA: uncorrected intermediate visual acuity.

**Table 4 life-15-00064-t004:** Visual satisfaction questionnaire CatQuest-9SF results pre- and post-bilateral cataract surgery using mini-monovision technique in eyes implanted with Clareon intraocular lens.

Questions	1: Very Dissatisfied		2: Quite Dissatisfied		3: Quite Satisfied		4: Very Satisfied		5: I Can’t Decide	
	Pre	Post	Pre	Post	Pre	Post	Pre	Post	Pre	Post
A—Difficulties in any way in daily life	50%	0%	33.3%	0%	16.6%	16.6%	0%	83.3%	0%	0%
B—Satisfaction with vision/sight										
Reading text in newspaper	50%	0%	50%	0%	0%	16.6%	0%	83.3%	0%	0%
Recognizing faces of people you meet	16.6%	0%	66.6%	0%	16.6%	23.3%	0%	76.6%	0%	0%
Seeing prices of shopping goods	0%	0%	50%	0%	50%	10%	0%	90%	0%	0%
Seeing to walk on uneven ground	33.3%	0%	50%	0%	16.6%	33.3%	0%	66.6%	0%	0%
Seeing to perform needlework or handicrafts	50%	0%	50%	0%	0%	33.3%	0%	66.6%	0%	0%
Reading subtitles on the TV	50%	0%	33.33%	0%	16.6%	33.3%	0%	66.6%	0%	0%
Seeing to carry out a preferred hobby	33.3%	0%	50%	0%	16.6%	73.3%	0%	26.6%	0%	0%

## Data Availability

The dataset of the study is available in this public repository: https://zenodo.org/records/13716481.

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
