# Peer review of "Visual Performance After Bilateral Implantation of a New Enhanced Monofocal Hydrophobic Acrylic Intraocular Lens Targeted for Mini-Monovision"

_life, 2025, doi:10.3390/life15010064_

Round 1
Reviewer 1 Report
Comments and Suggestions for Authors
If there is a specific reason for not measuring near binocular vision, I would appreciate it if you could explain. The abstract contains extensive details about the survey. It would be better to summarize and include only the essential information. Additionally, including the results of the control group that did not undergo mini-monovision would also be necessary. It is essential to include content that highlights the differences between the existing AcrySof IOL intraocular lenses and the ones being discussed. It would be helpful to include the monocular results alongside the defocus curve for a more comprehensive comparison. For Table 4, it might be beneficial to reorganize it in a way that is easier for the general readers to understand. Simplifying the layout and presentation could improve accessibility and clarity. It would be helpful to include the results of the sample size calculation in the statistical analysis section for added clarity and rigor. Finally, it is necessary to verify whether changes in pupil size and the resulting higher-order aberrations were examined.
Author Response
1. If there is a specific reason for not measuring near binocular vision, I would appreciate it if you could explain.
Thank you for your comment. Indeed, near vision was evaluated at the time of performing the defocus curve. However, lenses evaluated in this study are monofocal and the programmed mini-monovision technique was mainly designed to obtain adequate visual performance at far and intermediate distances. Therefore, to meet the research objective, our evaluation was specifically oriented to measure far and intermediate visual performance, and even so, the desired refraction for the non-dominant eye was preoperatively programmed, as explained in the methods section.
2. The abstract contains extensive details about the survey. It would be better to summarize and include only the essential information.
We have simplified the summary, we hope to have fulfilled your request.
3. Additionally, including the results of the control group that did not undergo mini-monovision would also be necessary.
We agree with you. Thanks to your comment we have included it and emphasized as a limitation of the present study. Our study design was oriented to know the performance of this lens in a series of cases, performed in a frequent clinical context, using this monovision technique and we wish to share it, in order to justify future case-control studies. You can find the modifications we have made thanks to your comments between lines 251 to 263 of the discussion.
3. It is essential to include content that highlights the differences between the existing AcrySof IOL intraocular lenses and the ones being discussed.
Following your recommendation, we have included in the discussion a paragraph related to this (lines 230 to 238) Also, one more reference was added. The new reference is number 17: Werner L, Thatthamla I, Ong M, Schatz H, Garcia-Gonzalez M, Gros-Otero J, Cañones-Zafra R, Teus MA. Evaluation of clarity characteristics in a new hydrophobic acrylic IOL in comparison to commercially available IOLs. J Cataract Refract Surg. 2019;45(10):1490-1497. doi: 10.1016/j.jcrs.2019.05.017.
4. It would be helpful to include the monocular results alongside the defocus curve for a more comprehensive comparison.
Dear reviewer, unfortunately we have not performed the monocular defocus curves since we have focused on binocular vision obtained with the mini-monovision technique. We hope you understand us. Performing them would have tripled the evaluation time with each patient and would not have provided us with much more information, according to the proposed objective.
5. For Table 4, it might be beneficial to reorganize it in a way that is easier for the general readers to understand. Simplifying the layout and presentation could improve accessibility and clarity.
Thank you, we have redesigned table 4.
6. It would be helpful to include the results of the sample size calculation in the statistical analysis section for added clarity and rigor.
Thank you, we have included this information in the statistics section.
7. Finally, it is necessary to verify whether changes in pupil size and the resulting higher-order aberrations were examined.
We have completed this information in the methods section on lines 93 and 94.
More changes
*Following the recommendations of another reviewer, we have added an extra paragraph in the discussion, between lines 271 and 286.
**Finally, we would like to thank you for allowing us to improve the content of our study based on your comments. We hope that this new version of the study will meet your requirements.

Reviewer 2 Report
Comments and Suggestions for Authors
I thank the Editor for the opportunity to review this manuscript. The paper is interesting and well written. Here follow my comments.
1) Limitations are not listed in the paper. The main limitation of the paper is the lack of a control group. This would have allowed for a direct comparison between patients in which emmetropia was targeted in both eyes, and patients with mini-monovision with Clareon IOLs, allowing to better assess the advantages to aim for mini-monovision over bilateral emmetropia.
2) There is no mention about contrast sensitivity tests at different pupil size. This should be mentioned among limitations.
3) Line 107: “The spherical equivalent of residual myopia was calculated between -0.25 and +0.25D for the dominant eye”. From 0 to +0.25 is not a myopic target but an hypoeropic one, please correct.
4) The authors used the CatQuest-9SF questionnaire as validated PROM. As they report, 66% and 90% of the patients reported to be “very satisfied” when doing needlework, handicraft and when reading price tags, respectively. This is not an intermediate vision task but a near vision task. In the defocus curve binocular UNVA is reported to be around 0.74 LogMAR for 40 cm, not enough to allow a satisfying UNVA especially when sewing. Maybe these 2 results refer to CNVA? I suggest to comment these findings.
5) Line 90 the word “Tomry” in the brackets should be corrected to Tomey”
6) Line 195: a square bracket is missing where a “*” was placed.
Author Response
Reviewer Responses
I thank the Editor for the opportunity to review this manuscript. The paper is interesting and well written. Here follow my comments.
Dear reviewer, we appreciate the time you have taken and the suggestions you have given us, with which we believe that our work has been improved. We have considered all your comments, which have been included in our new version. Once again, thank you very much.
1) Limitations are not listed in the paper. The main limitation of the paper is the lack of a control group. This would have allowed for a direct comparison between patients in which emmetropia was targeted in both eyes, and patients with mini-monovision with Clareon IOLs, allowing to better assess the advantages to aim for mini-monovision over bilateral emmetropia.
Thank you very much for highlighting this aspect. We agree with you, so in our new version we have elaborated our study limitations. You can find it in lines 251 to 263 of the discussion.
2) There is no mention about contrast sensitivity tests at different pupil size. This should be mentioned among limitations.
Thank you for your comments. It has been added to the discussion where we express our limitations.
3) Line 107: “The spherical equivalent of residual myopia was calculated between -0.25 and +0.25D for the dominant eye”. From 0 to +0.25 is not a myopic target but an hypoeropic one, please correct.
We agree. We have modified the sentence by deleting the words “residual myopic”.
4) The authors used the CatQuest-9SF questionnaire as validated PROM. As they report, 66% and 90% of the patients reported to be “very satisfied” when doing needlework, handicraft and when reading price tags, respectively. This is not an intermediate vision task but a near vision task. In the defocus curve binocular UNVA is reported to be around 0.74 LogMAR for 40 cm, not enough to allow a satisfying UNVA especially when sewing. Maybe these 2 results refer to CNVA? I suggest to comment these findings.
This is a very interesting comment. This has made us review the original data and we can confirm that the patient responses have been without wearing glasses. But we agree with you that sewing is a demanding near vision task and the best visual performance achieved in our series is in intermediate and not near vision. This controversy stimulated us to add one more paragraph to our discussion, in order to refer to the issue that the results of questionnaires should be taken with caution, since we do not really know how patients achieved these activities: for example, we think that possibly our patients to sew or perform a handicraft have moved away the object until they have it at an adequate visual distance, which certainly in our case will have been about 60 to 70 cms.
Thank you, your comments have allowed us to go deeper into this subject. You will find the development in our new version, between lines 271 and 286.
5) Line 90 the word “Tomry” in the brackets should be corrected to Tomey”
Thank you, it has been corrected.
6) Line 195: a square bracket is missing where a “*” was placed.
Thank you, it has been corrected.
More changes
*Dear reviewer, we have redesigned table 4 in response to another reviewer's request. Only a redesign was done and no result data was changed; we hope it will now be simpler to visualize.
**We have included in the discussion a paragraph regarding difference between Acrysoft and Clareon (lines 230 to 238) Also, one more reference was added. The new reference is number 17: Werner L, Thatthamla I, Ong M, Schatz H, Garcia-Gonzalez M, Gros-Otero J, Cañones-Zafra R, Teus MA. Evaluation of clarity characteristics in a new hydrophobic acrylic IOL in comparison to commercially available IOLs. J Cataract Refract Surg. 2019;45(10):1490-1497. doi: 10.1016/j.jcrs.2019.05.017.
Round 2
Reviewer 1 Report
Comments and Suggestions for Authors
I am wondering if you have examined the changes in spherical aberration values after the surgery. Since this is a retrospective study, please calculate and quantify the statistical power based on the current sample size and include it in the manuscript.
Author Response
Comment reviewer 2
I am wondering if you have examined the changes in spherical aberration values after the surgery. Since this is a retrospective study, please calculate and quantify the statistical power based on the current sample size and include it in the manuscript.
__________________________________________________________________________
Dear reviewer, again thank you for your time and for making us consider how to improve our work. We have separated your comments into two parts.
Regarding the first one, about assessing changes in spherical aberration after surgery, the answer is no. In our research protocol, we consider the parameter of spherical aberrations as preoperative demographic data, just as one more data to know and characterize the population to be operated.
But your question made us wonder about the usefulness of this data and whether we could have taken more advantage of it by analyzing it postoperatively. The value we analyzed, the corneal spherical aberration (Q factor), changes mainly in response to actions or modifications occurring in the cornea. Therefore, in the future, we believe that the ideal would have been to evaluate not only the corneal aberrations but also the total aberrations of the eye, in order to really know mainly the changes induced by the intraocular lens.
We take your suggestion, we regret not being able to have what you requested, since even if we recited all the patients to make the postoperative measurement of the corneal aberration, we do not believe we can obtain a relevant data for the present study. But on this subject and based on your comments, we have added a new text in the disucusion, in the limitations section (lines 527 to 562).
Thank you.
Regarding the second aspect (the second sentence of your comment), the design of our study, as expressed in the methods section, has been prospective. We have likewise added requested information on the statistical power of the current sample.
Once again we express our gratitude.
Reviewer 2 Report
Comments and Suggestions for Authors
The Authors fully replied to my comments. The paper can now be considered for publication
Author Response
Dear reviewer, thank you very much for your suggestions and for allowing us to improve.